# An integrated data framework for policy guidance during the coronavirus pandemic: Towards real-time decision support for economic policymakers

**Julian Oliver Dörr** [1,2☯] *, **Jan Kinne** [1,3☯], **David Lenz** [2,3☯], **Georg Licht** [1‡], **Peter Winker** [2‡]

**1** Department of Economics of Innovation and Industrial Dynamics, ZEW – Leibniz Centre for European Economic Research, Mannheim, Germany, **2** Department of Econometrics and Statistics, Justus Liebig University Giessen, Gießen, Germany, **3** istari.ai, Mannheim, Germany

☯ These authors contributed equally to this work.
‡ These authors also contributed equally to this work.
* julian.doerr@zew.de

**Data Availability Statement:** This work uses firm-level survey and credit rating data which cannot be published. The data is available at the ZEW Data Research Centre, http://kooperationen.zew.de/en/

## Abstract

Usually, official and survey-based statistics guide policymakers in their choice of response instruments to economic crises. However, in an early phase, after a sudden and unforeseen shock has caused unexpected and fast-changing dynamics, data from traditional statistics are only available with non-negligible time delays. This leaves policymakers uncertain about how to most effectively manage their economic countermeasures to support businesses, especially when they need to respond quickly, as in the COVID-19 pandemic. Given this information deficit, we propose a framework that guided policymakers throughout all stages of this unforeseen economic shock by providing timely and reliable sources of firm-level data as a basis to make informed policy decisions. We do so by combining early stage 'ad hoc' web analyses, 'follow-up' business surveys, and 'retrospective' analyses of firm outcomes. A particular focus of our framework is on assessing the early effects of the pandemic, using highly dynamic and large-scale data from corporate websites. Most notably, we show that textual references to the coronavirus pandemic published on a large sample of company websites and state-of-the-art text analysis methods allowed to capture the heterogeneity of the pandemic's effects at a very early stage and entailed a leading indication on later movements in firm credit ratings. While the proposed framework is specific to the COVID-19 pandemic, the integration of results obtained from real-time online sources in the design of subsequent surveys and their value in forecasting firm-level outcomes typically targeted by policy measures, is a first step towards a more timely and holistic approach for policy guidance in times of economic shocks.

## 1 Introduction

COVID-19 and its economic consequences have placed numerous firms under severe distress. In almost all countries, stores and businesses were closed and mobility severely restricted to contain the spread of the virus. While these large-scale anti-contagion policies had provably

zew-fdz upon request. The website data is archived in ZEW's Dark Archive for the purpose of adherence to scientific standards and proving correct scientific work according to § 60d III p. 2 of Copy Right Act, https://www.zew.de/en/das-zew/serviceeinheiten/zentraledienstleistungen/bibliothek/. A current version of the website data may be collected using the ARGUS Webscraper (https://github.com/datawizard1337/ARGUS). The code for analyzing the data as presented in the study is available at https://github.com/julienOlivier3/DataFramework_EconomicCrises.

**Funding:** The study extends upon a project analyzing the economic effects on SMEs in the COVID-19 crisis which was funded by the German Federal Ministry of Economic Affairs and Energy (https://www.bmwi.de) under the grant agreement No 15/20 (GL). Funding covered the collection of webdata and the analysis of both webdata and survey data. Moreover, the project received support by the Ministry of Science, Research and the Arts of the government of Baden Wuerttemberg (https://mwk.baden-wuerttemberg.de) as part of its Science Data Center program under the grant "Business and Economic Research Data Center (BERD)" (GL). The funders had no role in study design, data collection and analysis, decision to publish, or preparation of the manuscript.

**Competing interests:** The authors have declared that no competing interests exist.

positive effects on health outcomes [1], they fundamentally changed the landscape for many businesses. Due to the forced halt of many economic activities and the severe shock to global trade, many companies faced a situation of reduced business activity and declining sales figures, as well as major disturbances to their value chains and supplier networks, which had immediate consequences on the affected firms' financial positions.

The impact of COVID-19 on businesses has shown, however, a great degree of heterogeneity [2–4]. In some sectors firms have been barely affected by the pandemic or have even benefited from it, while in others large numbers of companies have been pushed into financial distress. Besides sector-specific differences, the economic exposure to the pandemic has also strongly varied with companies' business models. Some operations managed to adjust swiftly to the changed conditions, others had little scope to do so [5].

Now, after more than a year of pandemic, the winners and losers of the crisis seem rather clear. While firms with highly digitized business models such as delivery companies, e-commerce as well as online video conferencing and education platforms have thrived, companies whose business models are characterized by physical human interaction such as culture, travel, hospitality, restaurants and retail trade have greatly suffered [3]. What seems clear today, has however not been obvious at an early stage of the shock, when policymakers were confronted with various forms of economic uncertainty [6, 7] and stepped largely in the dark about the impacts of the pandemic on different businesses and different industries. Not only the dynamics of the pandemic were hard to foresee at an early stage of the crisis, but also governments' economic response measures have been unprecedented such that referencing to previous experiences has neither been useful nor possible. A dilemma for policymakers, who were forced to act quickly to cushion the economic impact of their virus containment measures, which severely added to the plight of businesses.

In fact, the shutdown measures not only required companies to reorganize their operations by adjusting to the changed conditions, but also led to a fast erosion of equity positions among heavily exposed companies. This brought many firms on the brink of financial solvency [8] and thus called for fast government assistance [9]. Faced with the threat of a wave of corporate insolvencies and its immediate consequences, such as mass layoffs, but also with a lack of information about the heterogeneity of the economic impact of the shock, policymakers granted liquidity subsidies and other support instruments on an unprecedented scale [9]. In Germany, the focus country of our study, for example, the government even launched the 'largest assistance package in the history of the Federal Republic of Germany' [10, pg. 3] comprising public net borrowing of around €156bn [10].

The lack of early indicators, signaling which firms were at highest risk to suffer liquidity shortfalls and indicating how sectors and regions were differently exposed in the early stage after the economic shock [9], left policymakers uncertain about how to most effectively steer countermeasures. As a result, most of the early stimulus was awarded on a lump-sum basis, without taking into account that not all companies were equally affected by the pandemic [11] (e.g. in Germany, liquidity grants' 'application and payment process [needed] to be swift and free from red tape' according to the Ministry of Finance [12, para. 2]. In the context of public loan programs, 'the credit approval process [did] not involve additional credit risk assessment by the bank' and 'there [were] no requirements for collateral security' [13, para. 5]). In this sense, the coronavirus pandemic demonstrated that in highly dynamic times, policymakers face information deficits that leave them uncertain about how to most effectively manage countermeasures [14]. Especially if quick intervention is required to mitigate social costs, as in the early stage of the pandemic, policymakers had no option but to grant economic aid in a largely indiscriminate manner, often at high fiscal burden. Overcoming these information

deficits is therefore crucial for policymakers to steer their response measures more effectively into directions where help is needed most urgently while not overburdening fiscal budget.

Usually, policymakers draw their information from official and survey-based statistics to decide over economic stimulus measures. However, after a sudden and unforeseen shock caused incalculable and fast-changing dynamics, firm-level data from these traditional information sources is usually not yet available due to their rather inflexible and slow update cycle. A problem that applies in particular to information about smaller, unlisted companies [15]. Besides the lack in timeliness, surveys are also costly and typically do not serve well to study heterogeneity across regions and firm-specific subgroups given their relatively small sample sizes [16]. With no time to wait for official surveys to reveal the early effects of the sudden Corona shock, many governments have thus started to experiment with alternative real-time data sources during the pandemic to better understand its economic impact [14]. This has also called economic research, as important guide to public-sector decision-making, to integrate timelier sources of data at a granular level when consulting political decision-makers [17].

Following this call, we present an approach that tracks communication patterns on corporate websites to disambiguate the heterogeneity of the pandemic's impact at both industry and firm levels. Our approach relies on textual references to the coronavirus pandemic published by companies on their corporate websites. We refer to a coronavirus reference as self-reported text fragment (sentence or paragraph) that contains specific keywords associated to the pandemic and the SARS-CoV 2 virus. Our analyses show that companies used their websites to communicate about the pandemic in different contexts. Given the different context of the text references, it is possible to construct impact indicators that reflect in which dimension the firm is affected by the pandemic. State-of-the-art methods from the field of Natural Language Processing (NLP) allow to derive these indicators. We apply our framework to a large sample containing all economically active firms in Germany that have their own web domain. The dynamic nature of website data allowed us to provide policy-relevant insights at a very early stage and at near real-time, way before alternative sources could reveal first patterns at comparable granularity. Specifically, our results reveal strong heterogeneity, disaggregated by regions and at fine granular level of industry affiliation. Moreover, we show that the communication patterns serve as leading indicators for liquidity shortfalls at the firm-level. We argue that these early findings serve as an empirical guide for policy actors to initiate more targeted policies, in a situation where other data cannot yet provide information on the underlying dynamics.

We acknowledge that surveys and official data, despite their lack in timeliness, are nonetheless important instruments for designing medium-term to long-term responses. Therefore, we propose a broader data framework for policy guidance that incorporates data from such sources as they become available over the course of an unexpected shock. In our study, we integrate results from a consecutive questionnaire-based business survey as well as proprietary credit rating data to assist policymakers with deeper insights that go beyond the website-generated indicators. The idea of integrating early results obtained from real-time online sources in designing subsequent surveys bears the potential of a more timely and holistic approach for policy guidance that is generally applicable in times of economic shocks. This not only allows policymakers to react more swiftly and targeted but also enables the design of medium to long-term stimulus packages based on a rich set of information that has been continuously updated over all stages of the shock. In that sense, our framework focuses on bridging the information gap that arises when traditional data collection can only create policy guidance with non-negligible time delays, especially in such highly dynamic situations as the coronavirus pandemic.

The remainder of this paper is structured as follows. Section 2 provides an overview of the relevant literature. Section 3 introduces the different sources of firm-level data that we use to

capture the impacts of the COVID-19 shock on German businesses at different stages of the pandemic and at different levels of granularity. The section also discusses the insights that were generated from these sources. Section 4 empirically examines and highlights the value of webdata as source to generate early indicators that reflect the impact of COVID-19 on the corporate sector. Section 5 concludes.

## 2 Related literature

This study contributes to the fast growing literature on the economic effects of the coronavirus pandemic. Naturally, financial markets deliver very early expectation-based insights to what extent an exogenous shock such as COVID-19 affects the corporate sector. Ding et al. [2], for example, analyze the relationship between firm characteristics and financial market reactions using stock market information from January to May 2020 for a large number of internationally traded firms. They find that especially firms that were strongly exposed to international supply chains, with comparatively weak pre-crisis financial standing and with higher ownership by hedge funds underperformed in the months after the outbreak of the pandemic. Based on U.S. stock market returns, Ramelli et al. [18] also analyze stock market performance in response to the COVID-19 shock but more strongly focus on the timing of the effects. They also find that internationally oriented companies, which have been severely affected by disruptions in world trade and have been heavily dependent on the Chinese market, performed poorly, especially at the beginning of the shock in January 2020. At a later stage, stock market reactions started to increasingly penalize companies with thin financial reserves, with consumer services seen as the hardest hit sector.

Further studies based on business surveys find that firms' survival expectations show great heterogeneity across industries and strongly depend on expectations concerning the duration of the shock's repercussions [19]. Based on a business survey conducted between March 28, 2020 and April 04, 2020, Bartik et al. [19] find that estimated survival probabilities are particularly low in arts and entertainment, personal services, the restaurant industry and in tourism and lodging. Using the US Current Population Survey, Fairlie [15] find that major industries such as construction, restaurants, hotels, transportation and other personal services experienced strong declines in the amount of active business owners in April 2020 due to the COVID-19 shock.

Central to this paper is the question to which extent alternative sources of online data (here: foremost text data retrieved from corporate websites) and novel methods to turn this raw data into valuable information (here: methods from the field of NLP) may help policymakers to make informed and evidence-based decisions in otherwise uncertain environments. With increasing amounts of (often unstructured) data available, improved computational resources and substantial advances in analytical techniques, this question has gained importance in recent years. Athey [20], for instance, argues that there are clear limits as to how sources of 'big data' and supervised learning techniques are useful for policy guidance. This is because 'there are a number of gaps between making a prediction and making a [good] decision' [20, p.483]. The former is where data-driven models clearly thrive, the latter, however, is subject to more nuanced trade-offs which are often not encrypted in data but rather require human rationalization. Clearly, this is also true for the many policy decisions that needed to be made in response to the COVID-19 shock. Weighing between shutdown measures to contain the spread of the virus and the economic damage caused by these measures is clearly such a rationalization. Likewise, granting state aid in a whatever-it-takes fashion to prevent the risk of a wave of business failures, as well as possible windfall effects if aid measures go to non-viable firms or firms which would not have required state support, is another trade-off policymakers

were confronted with in the early phase of the pandemic. Arguably, no data-driven model could have predicted the corporate outcomes resulting from different policy decisions, thereby implicitly relieving politicians of active decision making. This study, however, explores how a non-traditional, large-scale data source can serve as valuable guide in politicians' decision-making process. This question is of particular concern in situations where traditional, policy-guiding sources of small data are not yet available but swift policy action is required. In this vein, we see our study as an important contribution to the discussion about the value of combining and supplementing small data sources such as surveys with large scale online sources. In the social sciences, the integration of such heterogeneous sources of information is considered to be high and has recently been documented in various studies (e.g. [21–24]). In this context, our paper contributes to a specific use case where the combination of small and big data sources allows overcoming information deficits to reduce the risk of policy errors.

Moreover, in fragile situations where social stability is at stake, the pandemic demonstrated that it is paramount for policymakers to ensure accountability and maintain public trust in their decision making processes. Among policymakers, this has led to an increasing demand for evidence-based decision making in the wake of the COVID-19 crisis [25]. In this context, our framework provided political decision-makers with empirical evidence to legitimize policy decisions in a situation of otherwise limited information.

Finally, this paper contributes to the literature that exploits webdata as useful information source to tackle research and policy issues. In fact, the use of various sources of webdata to collect timely and reliable information has gained traction in recent years. For example, webdata from social media platforms is used for event detection to get an up-to-date picture of the situation regarding major social events [26, 27] or natural disasters [28, 29]. Both applications have also policy relevance in terms of public security and crisis management. In the field of economic research, data from company websites have also proven to be a valuable information resource. Companies typically use their websites to report on their products and services, to present their activities and reference customers, but also to inform their customers and partners about current events related to their business activities [30, 31]. Using this form of data comes, however, with a number of requirements and challenges in terms of data acquisition, data analysis and data validation. The extraction of relevant information from unstructured or semi-structured text data from corporate websites can be seen as particularly challenging here. At the same time, it promises a number of benefits, particularly in terms of granularity, timeliness, scope and cost of collection [32]. These benefits will also turn out to be key in this study. In addition to simple keyword-based approaches, e.g. to measure the diffusion of standards [33], approaches with more sophisticated NLP and Machine Learning (ML) methods in particular, have been successfully used, to generate web-based firm-level innovation indicators [34, 35], for instance.

In the following section, we will introduce a three-stage framework to analyze the impacts of the COVID-19 pandemic on the corporate sector in Germany. Special attention is paid to the first 'ad hoc' stage of our framework, in which we examined early phase pandemic-related dynamics on corporate websites for a large sample of German firms at near real-time (see also Kinne et al. [36]).

## 3 Multi-stage framework for crisis impact monitoring

The framework presented in this section is based on a multi-stage process that aims to provide an up-to-date and complete picture of the business landscape during the course of the unforeseen economic shock triggered by the coronavirus pandemic. At each stage, heterogeneous sources of information are used to shed light on the pandemic-related dynamics and impacts

on the corporate sector. In a first 'ad hoc' stage, a monitoring system based on a systematic analysis of corporate websites is set up in the short run, which provides informative and up-to-date impact data at a very early phase and at near real-time right after the pandemic had hit the economy. In this first stage, we not only demonstrate how a dynamic stream of unstructured, digitized data can be used for the sake of updating political decision-makers when traditional forms of policy data is not available yet. We also show that indicators generated from this information source serve in forecasting how the pandemic has been materialized in the companies' performance. Based on the findings of the first stage, the second 'follow-up' stage focuses on surveys to highlight specific aspects of the crisis and their effects on businesses. Here, the knowledge gained in the first stage enables the design of more targeted surveys. In a final 'retrospective' stage, data that has only become available after the shock has materialized in the economy is used to determine the more structural impacts on firm outcomes. The objective of the proposed framework is to provide, first and foremost, decision support for economic policy. Thus, the third stage focuses on outcome variables that are typically targeted by policymakers. In this study, we focus on how the pandemic-related liquidity shortfalls materialized in companies' observed creditworthiness. Firms' creditworthiness is a key determinant for access to external funding, which was severely impaired for many companies as a result of the lockdown measures and therefore necessitated state-financed liquidity support.

A key focus of the proposed framework is timeliness of information to ensure that policymakers can justify their decisions on an empirical basis at every stage of an economic crisis. In a highly dynamic situation such as the coronavirus pandemic, where policymakers are forced to react swiftly, timeliness is a key criterion. That is why alternative sources of *timely* and *reliable* data for policy are particularly important in order to assist policymakers in designing ad hoc support measures. In this context, the idea to complement real-time online sources with traditional information sources provides a holistic approach to continuously update economic policymakers throughout all stages of any economic shock. Fig 1 gives a conceptual overview of the proposed framework and the data bases involved.

In the following, we will present the data, the methods as well as the impact results for all three stages of the framework. Finally, we outline how our proposed framework can be extended to be more generally applicable beyond the coronavirus case presented in this paper.

## 3.1 First stage: Ad hoc web-based impact analysis

Especially in the early weeks of the pandemic, the impact on and response of firms and in particular the heterogeneity across different economic subsectors and regions have been quite unclear until surveys and other official data revealed first insights. We filled this information gap by making use of 'COVID-19'-related announcements found on corporate websites. For this purpose, in the first stage of our framework, we have accessed corporate websites of about 1.18 million individual German companies from mid March 2020 to end of May 2020 twice a week and searched for references related to the pandemic. We have used micro data from the Mannheim Enterprise Panel (MUP) which contains information on all economically active German firms in late 2019 including their corporate web addresses [37]. Based on a labeled sample of these references, impact indicators that reflect in which context the companies reported about the pandemic have been modeled. This approach allowed to capture first patterns regarding the effects of the economic shock on corporations and its heterogeneity across different economic sectors. In the following, we will describe how we proceeded in capturing COVID-19 references from company websites and how we turned these text fragments into meaningful indicators.

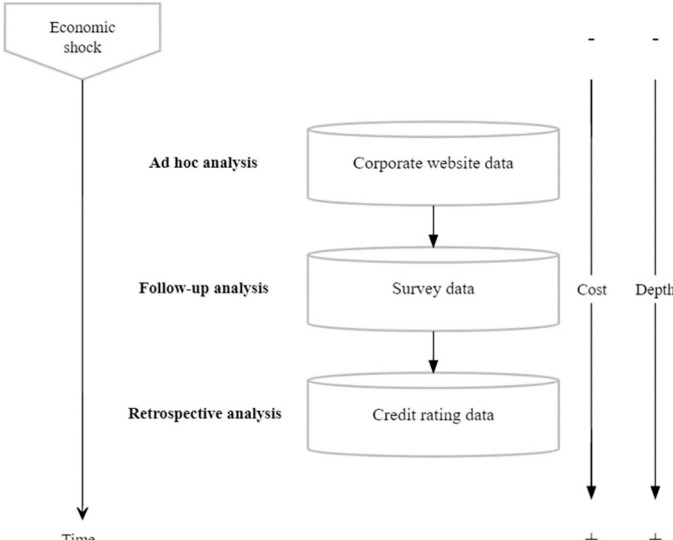

**Fig 1. Framework visualization.** Note: This figure illustrates the data framework for tracking effects of economic shocks on businesses by combining corporate website, business survey and credit rating data.

In a first step, the companies' websites were queried and downloaded following a structured approach. For each corporate website address, a maximum of five webpages per company (a website usually consists of several webpages) were crawled. The selection of these webpages was not conducted at random, but followed a clear heuristic: first, webpages with the shortest Uniform Resource Locator (URL) within the corporate website domain and whose content is written in German were selected (see Kinne et al. [32] for more details on the scraping framework). The former selection criteria satisfy that those webpages with more general and up-to-date ('top-level') information were downloaded with priority making it more likely that recent Corona references were captured by the search query. The downloaded webpages were then searched for variations of the term 'COVID-19' and relevant synonyms (see S1 Table for a list of these search terms). If any of the pre-defined search terms matched, the respective Hyper-Text Markup Language (HTML) node was retrieved for further processing. This simple approach allowed for a first estimation of the number of companies reporting about the Corona pandemic on their websites as displayed in Fig 2.

In total, we queried the large sample of 1.18 million corporate websites 13 times at regular intervals during the first weeks of the COVID-19 crisis in Germany. Fig 2 reveals that at this very early stage of the outbreak, just three days after the German Federal Government announced the first nationwide economic shutdown on March 16, 2020, more than 110,000 German companies had already mentioned COVID-19 on their websites. This comprises close to 10% of the overall corporate website addresses available to us (see S2 Table for a decomposition of detected Corona references across sectors and firm sizes; S3 and S4 Tables provide detailed information on sector and firm size definitions used in this study). The growth figures in Fig 2 (red line) also show that, especially at the beginning of the pandemic, shortly after the first shutdown in Germany, information on company websites posed a highly dynamic source of crisis-related data. Within just a few days, the number of companies with Corona references grew by double digits in percentage terms. These figures suggest that corporate website content offers great potential for learning how companies are affected by the pandemic and how they are dynamically coping with the changing economic reality.

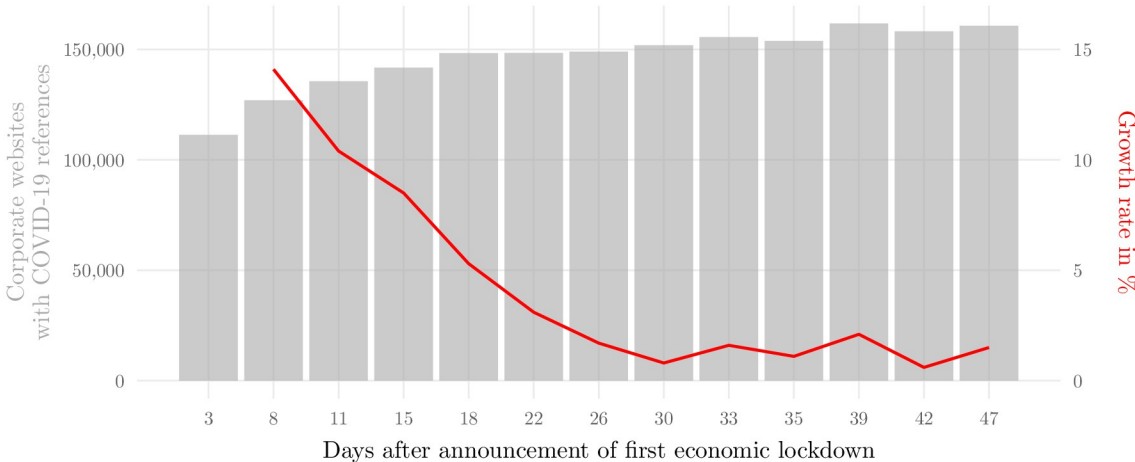

**Fig 2. Companies with COVID-19 references on their corporate websites after announcement of first economic shutdown.** Note: Figure shows the number of firms which reported about COVID-19 on their corporate websites over time, shortly after the announcement of the first nationwide shutdown at March 16, 2020 (left vertical axis). The repeated design of the web queries allowed to monitor the near real-time impact of the pandemic on the corporate sector. Red line (right vertical axis) depicts the growth rate of companies reporting about COVID-19 on their websites. Growth rate is calculated on a rolling basis with window size 3. Fluctuations towards the last few web queries both reflect an improved scraping process that was implemented in early May 2020 and companies that have removed COVID-19 references from their websites.

After the relevant text passages on company websites were identified, we continued in a second step with the classification of the context of the found Corona references. To this end, we introduced five different context classes (which we were able to identify after an exploratory analysis of the references). These context categories are defined as follows:

(1) **Problem**: The company reports on problems related to the Corona pandemic. This includes but is not exclusive to closures of stores, cancellations and postponements of events, reports of delivery bottlenecks and short-time work.

(2) **No problem**: The company reports that it is not affected by the Corona pandemic or that it has no impact on its operations.

(3) **Adaption**: The company reports that it is adapting to the new circumstances. This includes measures such as new hygiene regulations, changed opening hours, home office regulations and the like.

(4) **Information**: The company reports generally, not necessarily in a business-context, about the Corona pandemic. This comprises general information about the spread of the virus, symptoms of the disease, news about the pandemic or the announcement of official regulations.

(5) **Unclear**: This group includes texts that cannot be clearly assigned. Either they are artefacts or the reference does not come with further clearly distinguishable content.

In S5 Table, the interested reader finds examples for each of the five context categories.

With already more than 250,000 distinct 'Corona' references found in the first query wave in mid March, we have made use of a pre-trained language model from the transformers family [38] to scale the context classification task. Specifically, we adapted the XLM-RoBERTa architecture [39], a multilingual transformer model [38] pre-trained on over 100 languages. XLM-RoBERTa extends upon the seminal work on Bidirectional Encoder Representations

**Table 1. Distribution of context classes in the training data.**

|  | Problem | No problem | Adaption | Information | Unclear | Sum |
|---|---|---|---|---|---|---|
| Absolute | 1,007 | 241 | 1,441 | 750 | 834 | 4,273 |
| Relative | 0.236 | 0.056 | 0.337 | 0.176 | 0.195 | 1.0 |

from Transformers (BERT) [40] with an improved, robust pre-training. One advantage of the Transformer model class is that less training data is needed to achieve good classification results compared to text classification models that are trained from scratch. This is because transformer models are based on the concept of transfer learning. Transfer learning is a means to extract knowledge from a source setting and apply it to a different target setting. In the context of NLP, this means that the model is trained on large volumes of text data to learn general structure of language [41]. The general knowledge of human language structure that the model acquires during this pre-training phase, offers the benefit that much less—or even none, in the special case of zero-shot-learning [42]—training data is needed to adapt it to a new domain. Specifically, XLM-RoBERTa has been trained on more than two terabyte of filtered common-crawl data [39]. It has acquired its basic language understanding using the masked language model approach [40], i.e. given a sequence of text—e.g. a sentence—a random word is masked out and the training task is to predict the missing word. A pre-trained model such as XLM-Ro-BERTa can then be fine-tuned on a specific task. In our case, we fine-tuned XLM-RoBERTa to recognize the context of the retrieved COVID-19 references. Only this fine-tuning step requires labeled data that allows the model to adapt to a specific downstream task. For this purpose, we labeled a random sample of 4,273 of the retrieved text passages with their respective context class in order to build a training set.

As one can see from Table 1 the class distribution in the training data is rather unbalanced. Especially the 'no problem' category is underrepresented with less than 6% of total cases, while the 'adaption' class makes up around one third of the training data. The presence of unbalanced classes in the training data might lead to a sharp underestimation of the probability of rare events [43]. We therefore employ class weights inversely proportional to their respective frequencies in the training set.

$$w_j = \frac{N_{training}}{N_{classes} \cdot N_{training,j}} \tag{1}$$

with $N_{training}$ as the number of observations in the training set, $N_{classes}$ as the number of distinct context categories and $N_{training,j}$ as the number of training observations in class $j$. Weights computed according to this formula give higher weight to the minority classes and lower weight to the majority classes. During model training, the model parameter updates get multiplied by the class weights, thus giving stronger updates for less frequent classes and vice versa.

For the final context classification, we used an ensemble method [44], i.e. we trained multiple models on different subsamples of the training data. Model ensembles have been shown to increase robustness and decrease susceptibility to errors. The predictions of the individual models are aggregated and the final prediction is based on a majority vote.

The prediction performance of the trained model has been validated on test data consisting of an additional set of labeled website references that we did not use for fine-tuning the language model. For generating the test set, we engaged two independent annotators to manually assign 1,000 references to one of the five context categories. The *co*-annotation procedure allowed us to analyze how well the references can be distinguished based on the previously

**Table 2. Performance of context classification on test set.**

| Context classes | Precision | Recall | F1-score | Support |
|---|---|---|---|---|
| Problem | 0.99 | 0.98 | 0.98 | 290 |
| No problem | 1.00 | 0.22 | 0.36 | 18 |
| Adaption | 0.64 | 0.94 | 0.77 | 144 |
| Information | 0.66 | 0.95 | 0.78 | 117 |
| Unclear | 0.93 | 0.30 | 0.45 | 145 |
| Accuracy | | | 0.81 | 714 |

introduced context classes. For this purpose, we calculated Cohen's Kappa, $\kappa$, as conservative measure of inter-annotator agreement that controls for the possibility that annotators agreed by chance [45]. We find that the inter-annotator agreement is $\kappa = 0.62$ which can be considered as 'substantial' [46] and makes us confident that our context classes are distinguishable and serve well to capture the heterogeneity in the firm communication patterns. We then proceeded with the test references which both annotators assigned to the same context class (714 out of 1,000) to calculate the models classification performance. The performance metrics can be found in Table 2. They reveal several insights that we will discuss in the following. First, reports about pandemic-related problems are almost perfectly classified and retained by the model as can be seen by the 98% F1-score for the 'problem' context class. Next, the model retrieves with 94% (95%) a large fraction of reports about firms adapting to (informing about) the pandemic. If it classifies a website reference as adaption or information, this classification is correct in about 2 out of 3 cases. Finally, if the model predicts that a firm indicates having no problem in relation to COVID-19 (that the context of the reference is unclear), it is correct in all (93%) of the cases in the test set. However, the model retrieves only a small part of those references which were labeled as 'no problem' ('unclear') by the annotators, as shown by the 22% (30%) Recall. The overall accuracy of the model is 81%. The model's capability to almost perfectly predict and retain reports of firms facing pandemic-related problems is of particular importance in the subsequent regression analyses. There we show that the 'problem' category closely mimics results obtained from a business survey (see Fig 6) and that it serves as a robust leading indicator of firms' deterioration in their credit standing (see Table 7, column (4)).

We then used the trained model to classify all of the Corona references that we retrieved from the company websites. Table 3 provides descriptive statistics for the out-of-sample web references aggregated at the firm-level. For this purpose, we present the categories in a binarized version where the context class for firm $i$ equals 1 if the firm has reported on its website about COVID-19 in the respective context in any of our web queries. Otherwise, the respective context class for firm $i$ equals 0. It is worth noting that a firm can report about the coronavirus at several passages and in different contexts on its website. For this reason, the firm-level class assignments are non-exclusive. Descriptive statistics show that overall 17% ($N = 202,076$) of all German companies with a corporate website reported about the pandemic in some context. We calculate context-specific impact values, defined as the number of firms within the context category relative to the total number of 202,076 firms that reported about the pandemic. This is a simple yet effective measure to disentangle heterogeneity across different firm characteristics such as sector affiliation as demonstrated in the subsequent analyses (see Fig 3 for instance). At the aggregate level, the impact values reveal that 63% of the firms which reported about the coronavirus on their website, did so by mentioning adaption to the new economic circumstances. More than ⅓ ($N = 69,962$) of the companies with COVID-19 references

**Table 3. Descriptive statistics: Corporate website data.**

| Context classes | Fraction | Impact value | N |
|---|---|---|---|
| Problem | 0.06 | 0.35 | 69,962 |
| No problem | 0.01 | 0.06 | 13,118 |
| Adaption | 0.11 | 0.63 | 128,140 |
| Information | 0.05 | 0.31 | 62,174 |
| Unclear | 0.08 | 0.49 | 98,156 |
| Overall | 0.17 | | 202,076 |

Note: If a firm has reported at least one COVID-19 reference in any of the query waves that has been classified in the respective category, the firm gets assigned a 1. Else the firm gets assigned a 0 for the respective category (binarized version of the web indicators). The column 'Fraction' indicates the fraction of firms from the overall sample of 1.18 million websites that reported about the pandemic in the respective context category. Based on those firms with at least one COVID-19 reference, column 'Impact value' reflects the share of firms with references in the respective context category. *N* refers to the absolute number of firms with references in the respective context category. The 'Overall' row shows the overall number of firms with at least one COVID-19 reference both in relative terms (Fraction) and absolute terms (*N*). Note that the assignment to the context classes is non-exclusive at the firm-level since a company can report about the pandemic at several passages and in different contexts on its website.

signaled problems related to the pandemic and only a comparatively small number of 13,118 companies signaled the contrary of no problems.

A major advantage of this early assessment of communication patterns is that it allows to monitor impact variation across firm characteristics such as sector affiliation, age and geographic subgroups. In a situation of fast-changing dynamics and unforeseeable consequences this may help as important decision support for policymakers which otherwise have no other empirical basis to rely on. In what follows, we will demonstrate how the web-based indicators allow to uncover the heterogeneity of the pandemic's impact. In doing so, we present impact values disaggregated across different firm characteristics.

Fig 3 provides an overview how communication about the pandemic differs across industry sectors. Values in red represent sector-specific impact scores, defined as the proportion of companies that communicated about the pandemic in the respective context within that sector. Grey shaded areas represent the same indicator as unweighted average across all sectors. Most remarkably, the analysis clearly reveals disproportionately strong reporting of problems among firms in the accommodation & catering sector, where 58.02% of companies signaled facing issues due to the pandemic, and the creative industry & entertainment sector where this number even reached 77.47%. In contrast, less than 20% of the firms in the sectors chemicals & pharmaceuticals, insurance & banking, manufacturing of data processing equipment and mechanical engineering reported about pandemic-related problems. This clearly gives an early indication that heterogeneity of the crisis impact is substantial and that policy support in the most adversely affected sectors appeared most urgent. In other sectors, such as business-related services, insurance & banking and health & social services, firms relatively often only informed in a broader context about the pandemic on their websites which seems intuitive, especially in the latter case. Finally, it is interesting to see that in the insurance & banking sector a relatively large fraction of 21.18% of firms signaled that they are not negatively impacted by the economic shock and that they are strongly adapting to the crisis. Deeper investigation of the references revealed that banks and insurance companies adapted to the crisis by streamlining and digitizing their services while signaling that customer support and service quality remains unaffected by these initiatives and the pandemic in general.

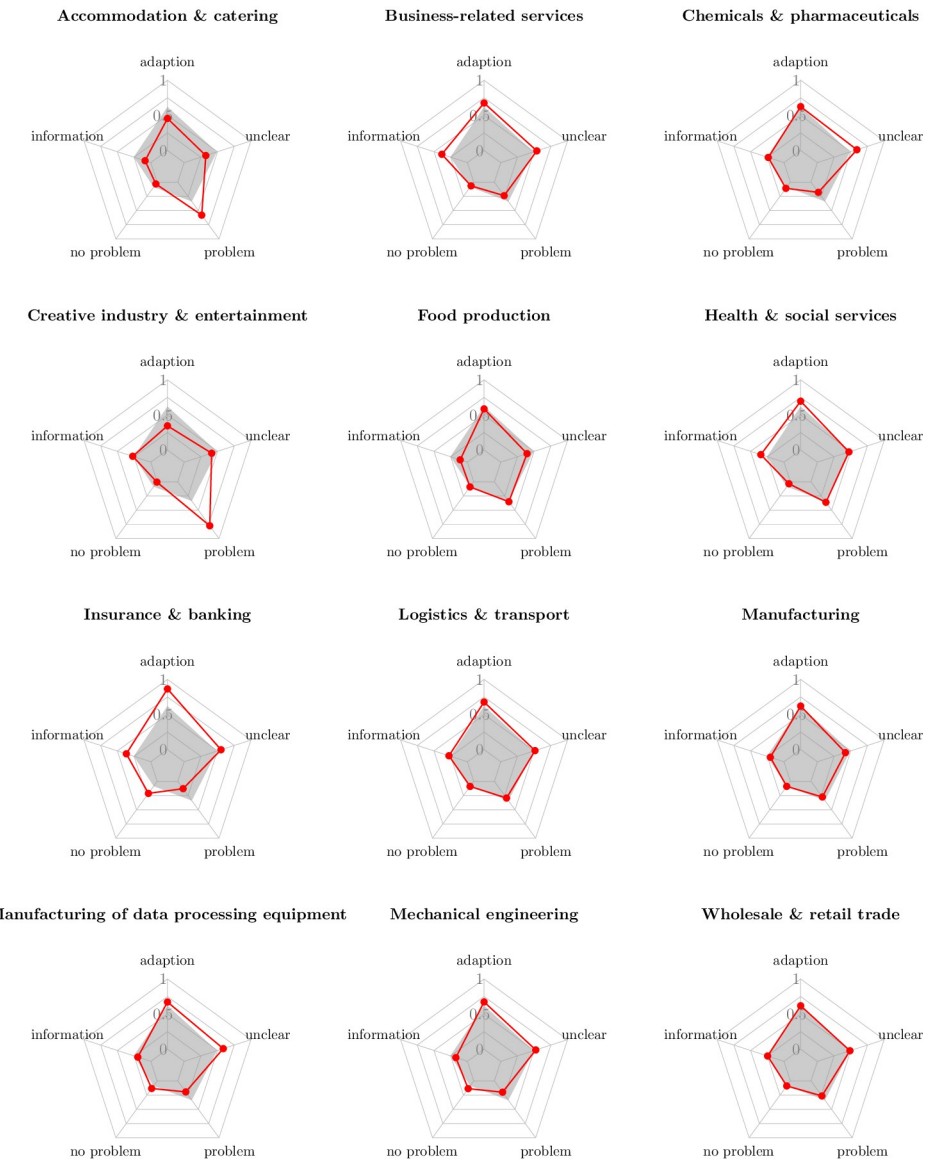

**Fig 3. COVID-19 firm communication on corporate websites website-generated impact values at sector level.**
Note: Visualizations based on classified COVID-19 web references. If a firm reported at least one COVID-19 reference
that has been classified in the respective context class in any of the web queries, the firm gets assigned a 1. Else the firm
gets assigned a 0 for the respective class (binarized version of the web indicators). Red lines represent sector-specific
impact values. Grey shaded areas represent unweighted average impact values across all sectors. Exact numerical values
can be found in S6 Table.

Given the large sample size of the webdata, these sectoral impacts can be further disaggre-
gated at a finer level of granularity. To demonstrate this, we focus exemplary on the heteroge-
neity of firms reporting about problems *within* the wholesale sector. The wholesale sector is
interesting for two reasons. First, the aggregated view on the trade sector, as displayed in the
lower right corner in Fig 3, does only reveal that around 26% of the firms communicated
about issues which is well below the (unweighted) average impact value of 34% across all sec-
tors. Survey-based data usually do not allow to break this insight further down to a subsector

level due to their relatively small sample size. Policymakers would thus be left with the information that negative impacts are below average for firms operating in trade, although some trade subsectors may be much more severely affected. Second, wholesale companies can provide important signals about the extent to which the national supply of certain goods and commodities may become tight. In the coronavirus pandemic, this has become a severe problem since international shutdown measures and changed consumer behavior have led to supply shortages of various goods. Problem reports in the wholesale sector disaggregated by single product markets, can help to signal the risk of supply issues at an early stage. While analyzing the relation between communication patterns and actual supply shortages is beyond the scope of this study, we still present the fine-granular impact values of distinct product markets within the wholesale sector. For this purpose, we follow the statistical classification of economic activities of the European Union [47] and assign all wholesale companies in our sample to close to 40 different product markets (see S7 Table for a detailed listing). From this disaggregation, we see that especially supply of household goods such as textiles but also manufacturing goods such as machineries, intermediates and related equipment were most adversely affected in the early stage of the pandemic. Basic supply such as food and beverages and, from a policy perspective also important, supply of pharmaceutical goods seemed to be less at stake since less than 15% of the respective wholesale companies reported about issues.

These near real-time insights into the heterogeneity of the Corona pandemic's impact on the business sector provide policymakers with a better understanding how early and more targeted impulse measures can be designed. Without the time to wait for official surveys to reveal the effects of the pandemic and shutdowns, we see this stage of our framework as an explorative analysis how governments can be assisted with empirical evidence from alternative data sources. While this section strongly focused on the heterogeneity across different industries and sectoral subgroups, it shall be clear that the presented approach can be used to set-up a monitoring system not only to track 'problem' sectors but also to unveil impact variation along further firm dimensions. For instance, the system can additionally account for regional differences based on the firms location (see, for example, [36] for an early version of the assessment of the Corona pandemic via corporate websites and S1 Fig).

## 3.2 Second stage: Follow-up survey-based effect differentiation

In a second stage, after firms have been exposed to the adverse economic environment for a critical period of time, we transfer our impact analysis from corporate website data to results obtained from a questionnaire-based survey which allows us to further differentiate the firm-level effects of the pandemic. The early insights from the first stage of our framework, serves here as valuable guide for the concrete design of the survey (e.g. formulation of questions, implementation of sampling strategy). For example, we could use the insights from the annotation and classification process in the first stage, to formulate relevant and specific survey questions concerning the types of problems companies were facing early on in the pandemic. This shows that the near real-time information obtained from the first stage of the framework allows for a more targeted design of follow-up surveys.

In order to guarantee a continuous update of policymakers information basis, the business survey has been conducted consecutively. Starting in mid of April, the survey has thus been repeated mid of June and end of September 2020. Over the course of the three survey waves, information on 1,478 distinct companies could be analyzed (the survey is a representative random sample of German companies, drawn from the MUP and stratified by firm size and industry affiliation—further details concerning sampling strategy and exact sample size for each of the survey waves can be found in S1 Appendix). Based on these consecutive surveys,

we analyze the different dimensions of the adverse impact of COVID-19 on businesses. Preparation and implementation of the survey required time and resources that only allowed to obtain these insights with a non-negligible time delay after first policy measures had already been implemented. The typical small sample size comes also with the obstacle that impact heterogeneity across fine granular sectoral subgroups cannot be disentangled. However, the advantage of the survey data is that it allows to capture the nature and extent of the negative impact of the pandemic on businesses in greater detail compared to the more timely assessment via website data. In other words, surveys provide an important addition of policy-guiding data in order to understand the various impact channels of an economic shock.

Table 4 shows how the design of the impact questions in the business survey enables a deeper understanding of the various effects of COVID-19 on the corporate sector. In question 1, companies were asked on a Yes-No basis whether they are generally negative affected by the COVID-19 pandemic. For a more nuanced understanding of the type of impact of the shock and the containment measures, firms were asked in a second set of questions, in which respect they were impacted on specific dimensions. These dimensions comprise (A) drop in demand, (B) temporary closing, (C) supply chain disruptions, (D) staffing shortages, (E) logistical sales problems and (F) liquidity shortfalls and were asked on 0–4 Lickert scale (0 indicates no negative effects, 4 signals strong negative effects). Descriptive statistics of the survey results in Table 4 show that 77% of the surveyed companies reported to be negatively affected by the pandemic at least in one of the three survey waves and that a drop in demand was on average the most severe problem among the six dimensions.

Fig 4 provides an overview how the exposure to the six impact dimensions differ across industry sectors. Similar to the impact analysis via corporate website data, the survey reveals disproportionately strong impacts in accommodation & catering and creative industry & entertainment. The survey results allow a more precise differentiation of the negative effects, which tend not to be published by the companies on their websites and are consequently hard to detect with a web-based analysis. In particular, a sharp decline in demand and temporary closure of business operations which are associated with a liquidity squeeze have placed hotels, restaurants, catering services, libraries, museums, operator of sports, amusement and recreation facilities as well as independent artists under severe distress. The forced halt of their business activities clearly justified public liquidity support, especially if the business models were running successfully before the outbreak of the pandemic. Sectors such as health & social services as well as manufacturing and engineering-related sectors show disproportionately strong exposure to the issue of supply chain disruptions and staffing shortages, but are barely

**Table 4. Descriptive statistics: Survey data.**

| Questions | Min | $Q_1$ | Median | Mean | $Q_3$ | Max | $N$ |
|---|---|---|---|---|---|---|---|
| 1: Overall-negative-impact | 0 | 0 | 1 | 0.77 | 1 | 1 | 1,478 |
| 2.A: Drop in demand | 0 | 1 | 2 | 2.14 | 4 | 4 | 1,176 |
| 2.B: Temporary closing | 0 | 0 | 0 | 1.19 | 2 | 4 | 1,278 |
| 2.C: Supply chain disruption | 0 | 0 | 0 | 1.04 | 2 | 4 | 1,202 |
| 2.D: Staffing shortage | 0 | 0 | 0 | 0.77 | 1 | 4 | 1,234 |
| 2.E: Logistical sales problems | 0 | 0 | 0 | 0.89 | 2 | 4 | 1,230 |
| 2.F: Liquidity shortfalls | 0 | 0 | 0 | 1.16 | 2 | 4 | 1,219 |

Note: Table shows descriptive statistics of survey questions. Values represent average values at the firm-level across the three survey waves. Question 1 is based on a Yes-No basis. Questions 2.A—2.F were asked on a 0–4 Lickert scale with 0 indicating no negative effects, 4 signaling strong negative effects. Non-responses in 2.A—2.F lead to lower observation numbers in these questions.

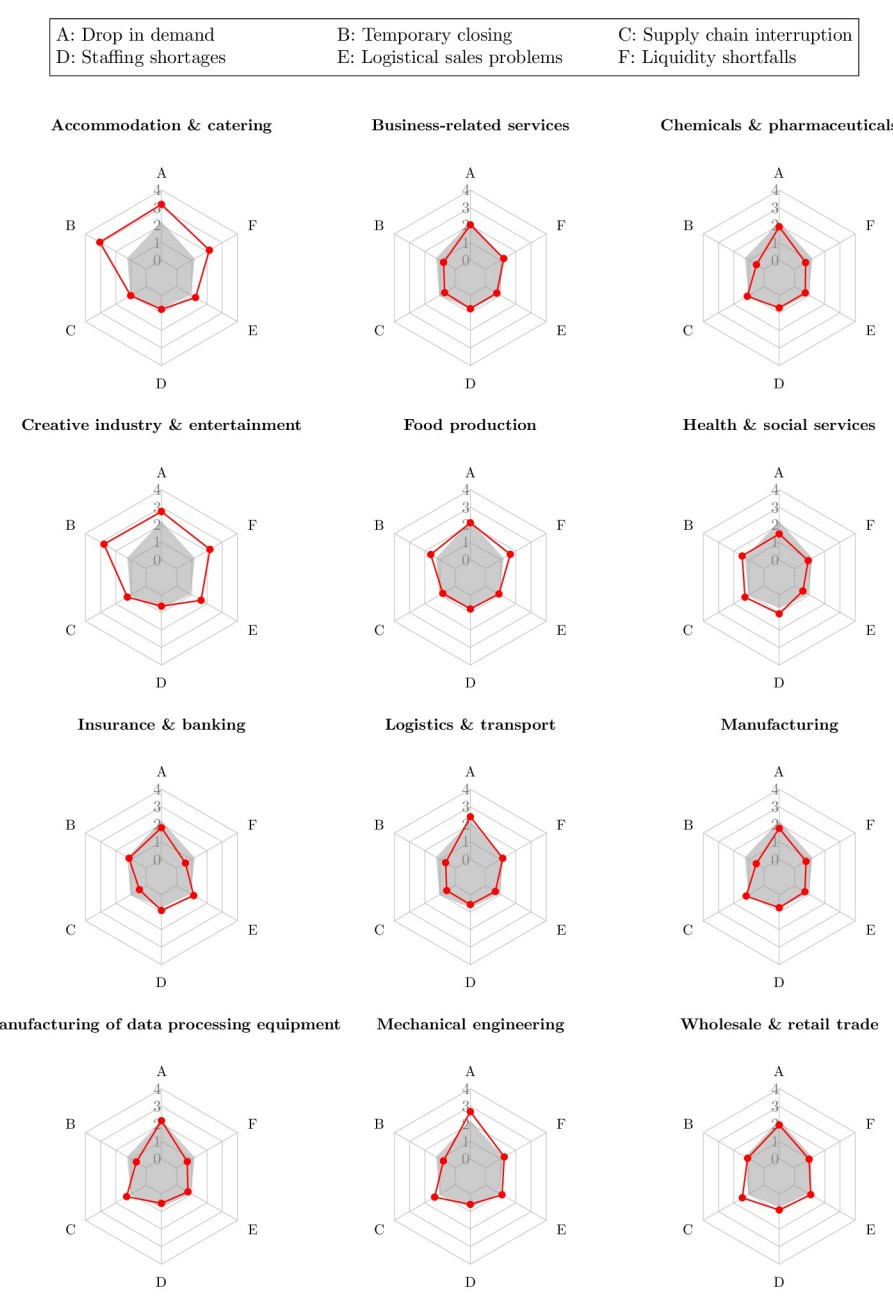

**Fig 4. COVID-19 firm exposure at sector level based on survey results.** Note: Visualizations based on survey questions A—F which were asked on a 0–4 Lickert scale with 0 indicating no negative effects, 4 signaling strong negative effects. Red lines represent sector-specific impact values. Grey shaded areas represent unweighted average impact values across all sectors. All values are averaged at the firm-level across the three survey waves.

confronted with declining demand numbers and liquidity shocks. It is clear that for firms in these sectors, the priority of policy should not be to provide liquidity support. An important issue for these firms is maintaining relationships with their stakeholders. Building these relationships is costly, and maintaining them despite the economic downturn is key to a successful recovery for many of these firms. Therefore, policymakers are challenged to renew and

reevaluate their policy toolkit to find new tools to help companies maintain their stakeholder relationships with workers and suppliers during economic downturns [9].

The second 'follow-up' stage of our proposed framework clearly demonstrated that, based on survey data, businesses in the accommodation, arts, and entertainment sectors have been facing strong liquidity bottlenecks, which in light of often unchanged fixed cost obligations poses a high risk of financial insolvency. In the third 'retrospective' stage of our framework, we more closely focus on this liquidation risk by analyzing the change in corporate solvency information in response to the COVID-19 crisis.

## 3.3 Third stage: Retrospective liquidation risk analysis

A major economic threat of COVID-19 has been and, given the possibility of recurring shutdown measures, continues to be, is the risk that firms with sound business models and decent financial performance before the outbreak of the pandemic are forced into insolvency. For economic policymakers it is important to understand if and where in the economy firms are at risk to leave the market permanently. Depending on size and strategic importance of impacted industries, this could imply high costs in terms of losses in jobs and output. In a third and last stage of our framework, we thus focus on this liquidation risk by transferring our impact analysis from corporate websites (first stage) and survey data (second stage) to firm-specific credit rating information which gives a much conciser picture to what extent the pandemic has materialized in the firms' financial solvency. For this purpose, we examine credit rating updates in the crisis period for more than 870,000 German companies. While firm-specific credit rating data reflect very precise information concerning the firm's financial standing and in case of substantial credit rating downgrades signals risk of financial insolvency [48, 49], the reassessment of firms' solvency by rating agencies is time and resource expensive. Generally, we find that on average the time between two credit evaluations equals 18 months. Typically, if credit information of a firm is requested more often by an external creditor, a company will be reevaluated more frequently. However, the rating capacity is largely tied to the headcount limitations of the rating agency. For this reason, only after a certain time a critical mass of rating updates becomes available to infer the heterogeneity of the crisis effects on companies' solvency.

The credit rating data that we analyze in the third stage of our framework is generated by Creditreform, Germany's leading credit agency. Creditreform assesses the creditworthiness of the near universe of active companies in Germany. The credit rating information is included for close to all firms in the MUP which allows to merge the ratings with the corporate website data (this becomes relevant in Section 4 of this study where we show how the webdata serves as leading indicator for later credit rating movements). Creditreform's corporate solvency index is based on a rich information set that closely mirrors a company's financial situation. Creditreform regularly investigates, among other things, information on the firm's payment discipline, its legal structure, credit evaluations of banks, caps in credit lines and further risk indicators based on the firm's financial accounts and incorporates this set of information into its rating score [50]. Different weights are attached to these metrics according to their importance for determining a firm's risk of defaulting on a loan. Overall, the rating index ranges from 100 to 500 with a higher index signaling a worse financial standing [51]. It is worth mentioning that the credit rating index suffers a discontinuity as, in case of a 'insufficient' creditworthiness, it takes on a value of 600. We truncate credit ratings of 600 to a value 500—the worst possible rating in our analysis. We do so since our main variable of interest is the *update* in the rating index which can only be reasonably calculated if the index has continuous support.

**Table 5. Descriptive statistics: Credit rating data.**

| Variables | Min | $Q_1$ | Median | Mean | $Q_3$ | Max |
|---|---|---|---|---|---|---|
| $\Delta r_t$ | -315 | -3 | 0 | 3.3 | 0 | 357 |
| Date of update | 2-Jun-20 | 3-Sep-20 | 30-Oct-20 | 21-Oct-20 | 8-Dec-20 | 9-Apr-21 |

Note: Table shows descriptive statistics of the rating updates and statistics of the dates of the rating revaluations. $\Delta r_t = 0$ means that the revaluation of the company has not led to any changes in its solvency compared to the pre-crisis period. $Q_1$ refers to the first quartile and $Q_3$ to the third quartile, respectively. The distribution of the dates of rating updates shows that more than 75% of the updates took place in 2020 and the latest update in the sample was conducted beginning of April 2021.

The level of the rating itself is little informative for inferring the effects of the COVID-19 crisis on the corporate sector. The change in the firms' credit rating, $\Delta r_t$, in contrast, precisely reflects to what extent a company has been down- or upgraded after the shock has hit the German economy. For that purpose, we consider all credit rating updates that have been conducted by Creditreform after June 1, 2020. We choose this date as it ensures that sufficient time has passed since the onset of the crisis to reflect COVID-related effects in the rating updates. The update in a firm's credit rating is defined as simple difference between the new rating index and the index before the update (i.e. before COVID-19) with a positive value indicating a downgrade and a negative value signaling an upgrade.

$$\Delta r_t = r_t - r_{t-x} \qquad (2)$$

Reassessments of the rating are conducted irregularly such that the time between two updates, $x$, varies. On average, the time between two updates in our sample equals 18 months.

Descriptive statistics in Table 5 show that most of the distribution is centered around 0, implying that a substantial number of firms experienced only minor changes in their credit ratings during the COVID-19 crisis. However, taking a closer look at the distribution of the rating updates across industry sectors in Fig 5 reveals an interesting pattern: sectors that, according to our first and second stage results, are severely affected such as logistics & transport, accommodation & catering and creative industry & entertainment but also supposedly winners of the crisis, most notably health & social services, follow a bimodal distribution. Comparing the crisis distribution with the pre-crisis distribution (indicated as dashed green line) suggests that this bimodality is indeed the result of the COVID-19 crisis. This means that major rating downgrades and upgrades are more likely in times of crisis than in normal times when a sector is severely affected by the crisis. We see this as strong hint that the pandemic shock has strongly materialized in severely affected firms' financial solvency with strong differences across sectors.

Moreover, the minimum and maximum values of $\Delta r_t$ in Table 5 show that there are some companies that have experienced large downgrades or upgrades in their credit ratings. To shed more light on this occurrence, Table 6 shows the fraction of firms with a substantial rating downgrade of more than 50 index points within the respective sector. We see again that logistics & transport, accommodation & catering and creative industry & entertainment show a relatively high fraction of firms which experienced a substantial downgrade in their ratings compared to less affected industries as well as compared to pre-crisis numbers. These high fractions of substantial rating downgrades reflect a relatively high insolvency risk in the respective industries. Despite the substantial policy support that these sectors received, this hints to a non-negligible number of market exits if support measures will cease before the firms have overcome the financial repercussions of the shock.

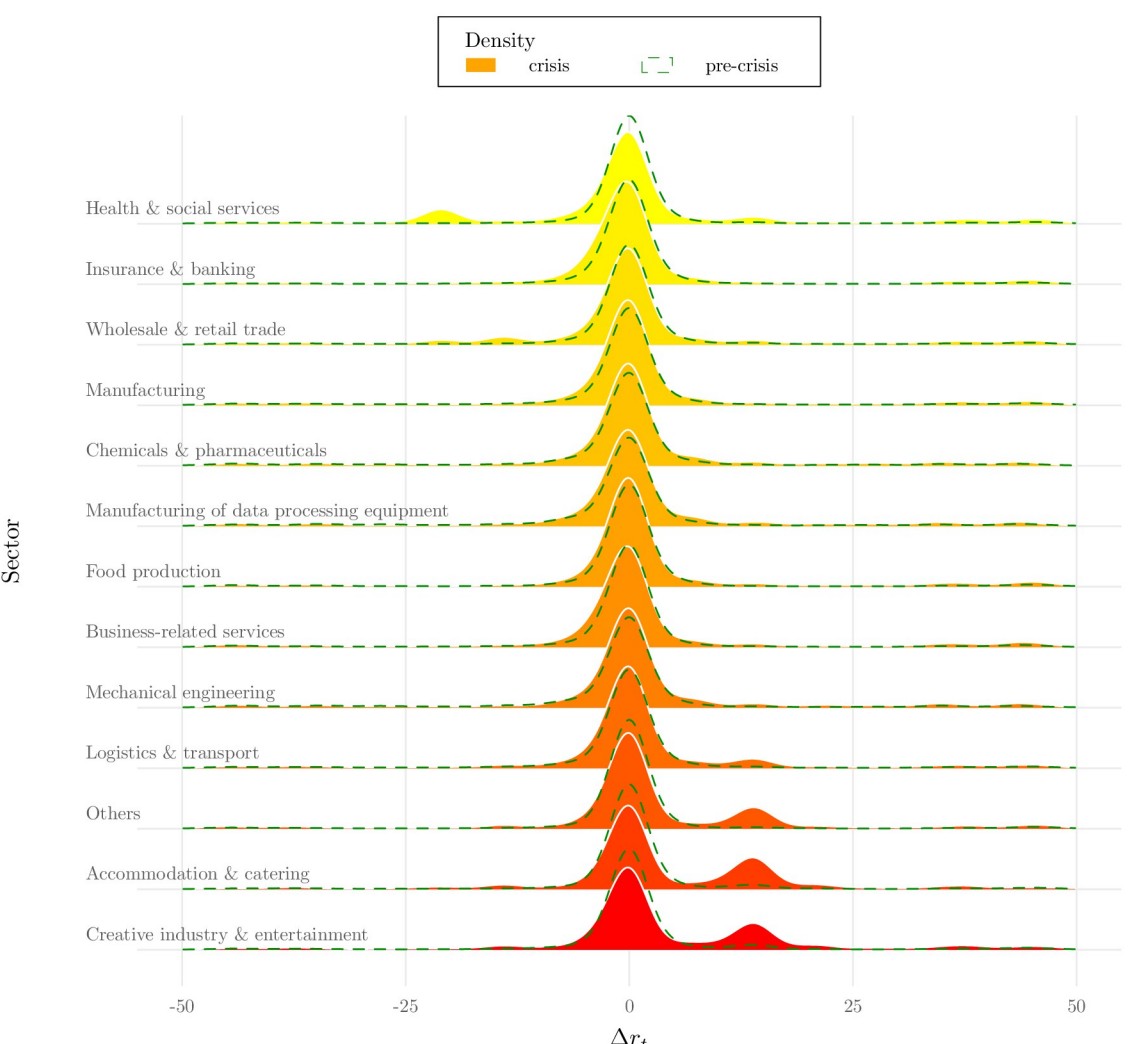

**Fig 5. COVID-19 effects on corporate solvency at sector level.** Note: Figure shows distribution of credit rating updates both during COVID-19 (yellow to red palette) and before COVID-19 (dashed green line). Densities are based on a Gaussian smoothing kernel with a bandwidth of 2.

In the last stage of our framework, we have focused on the structural risk of firms being forced to leave the market. Economic shutdown measures and drop in consumer demand have made this a particular concern of policymakers over the first months of the pandemic. Clearly, concerns about systematic bankruptcies have been specific to the COVID-19 pandemic and it is likely that the nature of future shocks will cause decision-makers to focus on different outcome variables. In the following section, we thus outline how our proposed framework can be extended to become applicable to a wider set of economic shocks and thus a useful tool to provide economic decision-makers with timely insights.

## 3.4 Generalizability to other types of shocks

While the approach presented to track the early impact of an economic shock using corporate communication patterns is specific to the coronavirus pandemic, the idea to integrate timely online sources with more traditional but less timely policy data can also be useful in other crisis

**Table 6. Distribution of extreme rating downgrades.**

| Sector | crisis | | pre-crisis | |
|---|---|---|---|---|
| | *N* | Substantial downgrades in % | *N* | Substantial downgrades in % |
| Insurance & banking | 34,768 | 3.0 | 35,087 | 1.7 |
| Manufacturing of data processing equipment | 4,512 | 3.1 | 4,406 | 2.7 |
| Chemicals & pharmaceuticals | 7,204 | 3.1 | 7,000 | 2.4 |
| Manufacturing | 224,813 | 3.5 | 204,613 | 2.7 |
| Food production | 10,420 | 3.8 | 10,311 | 2.9 |
| Health & social services | 62,633 | 4.0 | 57,466 | 2.0 |
| Mechanical engineering | 12,254 | 4.1 | 12,361 | 2.8 |
| Business-related services | 227,957 | 4.3 | 232,576 | 2.4 |
| Others | 14,259 | 4.8 | 12,511 | 2.0 |
| Wholesale & retail trade | 173,619 | 4.8 | 169,109 | 2.9 |
| Logistics & transport | 39,164 | 5.9 | 37,817 | 3.7 |
| Creative industry & entertainment | 13,865 | 8.7 | 12,967 | 3.9 |
| Accommodation & catering | 44,692 | 9.0 | 36,289 | 4.6 |
| Overall | 870,195 | 4.5 | 832,513 | 2.7 |

Note: Table shows fraction of firms with major credit rating downgrades by industry sector in percent. Substantial downgrades are defined as credit rating downgrades of more than 50 index points ($\Delta r_t > 50$). Pre-crisis numbers refer to the year 2018.

scenarios. Timely impact data from real-time online sources may not only reveal early heterogeneity at granular level or serve as leading indicators (as we will show in Section 4), but are equally important to design targeted surveys which in turn reveal more granular information how firms are affected by a shock and how and why possible heterogeneity in these effects may translate into heterogeneous firm outcomes. Integrating these different sources of information into a common framework allows policymakers not only to react more swiftly and targeted, but also allows to design medium to long-term stimulus packages based on a rich set of information that has been continuously updated over all stages of the shock. For example, one could think of immediate subsidies only for firms in hard-hit sectors (as evidenced by, e.g., a real-time online source) which suffered temporary liquidity constraints but are characterized by a robust pre-crisis performance (as indicated by, e.g., credit rating agencies and public annual reports). Moreover, policy decisions can be justified more easily if they are backed by empirical evidence. In this sense, the integration of real-time data sources for policy guidance is an important step towards evidence-based decision-making if unexpected dynamics require fast action. We suggest that this holistic approach to policy guidance, by combining different sources of information, bears the potential to be applicable to a wider range of economic shocks. As demonstrated in this paper, this requires a strategy of complementing specific crisis-related data sources. From early stage insights that are generated from real-time sources, to a follow-up stage based on targeted surveys, to a retrospective stage in the aftermath of the shock focusing on relevant outcome variables that have been identified in the earlier stages.

Ideally, a universally applicable system would be available to political decision-makers for this purpose in the future. However, a framework that can be used universally and on short notice requires that a large part of the early stage analysis is automated to a greater extent than in our study. This applies in particular to the selection of meaningful keywords, but also to the definition of 'impact' classes. Both could be done in the future, for example, by monitoring news streams. Business news articles could be used to filter relevant topics based on their

popularity and, for instance, a sentiment analysis. Relevant keywords could then be extracted automatically from all of these articles (based on unsupervised learning such as topic modeling whose topic-specific probability vectors over the vocabulary allow the identification of relevant keywords). These keywords would then be the input for a keyword search as described in our article, which would then be used as the basis for constructing classes (e.g., via clustering). Another promising approach for an extensive automation of our proposed framework could be the so-called zero shot classification. Zero shot does not require text analysis models to be fine-tuned for specific classifications, as is common in transfer learning. Instead, one relies on the general text understanding of the model learned in the original pre-training and works by posing each candidate label as a 'hypothesis' and the text sequence which we want to classify as the 'premise'. The zero shot model then estimates whether the hypothesis and premise match or not, respectively whether the assumption formulated in the hypothesis is confirmed by the text. Thus, without the time-consuming manual labeling of training data (as we have done in this paper), one can directly ask content-related questions with regard to individual sentences or text passages and, in the best case, receive a reliable answer.

Form this line of argumentation, it becomes apparent that more work should be dedicated in researching how real-time data sources can complement traditional forms of policy data, especially if empirical evidence is required for immediate government response. However, even if real-time data can be made accessible for policy guidance, an important question remains: What value do these information sources carry? In the next section, we show that the near real-time assessment via company communication patterns closely resembles heterogeneous effect estimates across various firm characteristics generated from survey responses. Moreover, we demonstrate that the webdata-generated impact values serve as leading indicators for companies' credit rating movements.

## 4 Assessing the predictive quality of early stage web-based impact indicators

The previous section has shown that all of the proposed data sources—corporate website data, survey data and credit rating data—hint to a strong degree of heterogeneity across economic sectors. While survey and credit rating data only revealed such patterns with a non-negligible time delay after the economic shock, corporate communication patterns retrieved from company websites indicated this heterogeneity at near real-time. A central question is to what extent the generated web indicators have predictive power in capturing the actual medium-term effects of the coronavirus shock. Clearly, predictive power is an important prerequisite for the web indicators to be useful for policymakers. Only if the webdata's early indication generates reliable insights, it bears the potential to help policymakers tailor their response measures and effectively channel economic assistance where it is needed most.

We assess the predictive value of the early web indicators by two distinct analyses: First, we compare the relationship between several firm characteristics and the negative shock exposure based on two identical regression specifications. The only difference between the two regressions is that we exchange the target variable, which in the first regression is generated from company website information (data from the first 'ad hoc' stage), while in the second regression it stems from the business survey (data from the second 'follow-up' stage). Second, based on a sub-sample of firms for which we have both COVID-19 web references as well as credit rating updates, we analyze to what extent the classified web references serve as leading indicators for later changes in the firms' credit rating.

To examine the statistical relationships between various firm characteristics and the negative effects of the COVID-19 shock on firms, we conduct a Probit regression. More precisely,

we regress a binary negative impact variable on age, size and sector characteristics.

$$Y_{k,i} = \alpha + \boldsymbol{\beta A}_i + \boldsymbol{\gamma S}_i + \boldsymbol{\delta I}_i + \epsilon_i \tag{3}$$

with

$$Y_{k,i} = \begin{cases} \text{problem}_i, & \text{if } k = \text{Webdata} \\ \text{overall} - \text{negative} - \text{impact}_i, & \text{if } k = \text{Survey} \end{cases}$$

and *A*, *S* and *I* matrices of company age, size and sector controls, respectively.

First, we conduct the regression estimation based on the corporate website observations. The dependent negative impact variable, $Y_{Webdata,i}$ equals 1 if the firm has reported a problem on its website and 0 otherwise. Second, we estimate the same regression specification based on the survey observations where the dependent variable reflects the first question in the survey: 'Has the coronavirus pandemic had negative economic effects on your company so far?' If the firm confirmed the question, $Y_{Survey,i}$ equals 1, otherwise it is 0.

Fig 6 visualizes the estimation results of both Probit regressions. Effect estimates need to be interpreted relative to the reference firm which is defined as an incumbent (10 years and older), micro company (less than 10 employees) in the accommodation and catering sector.

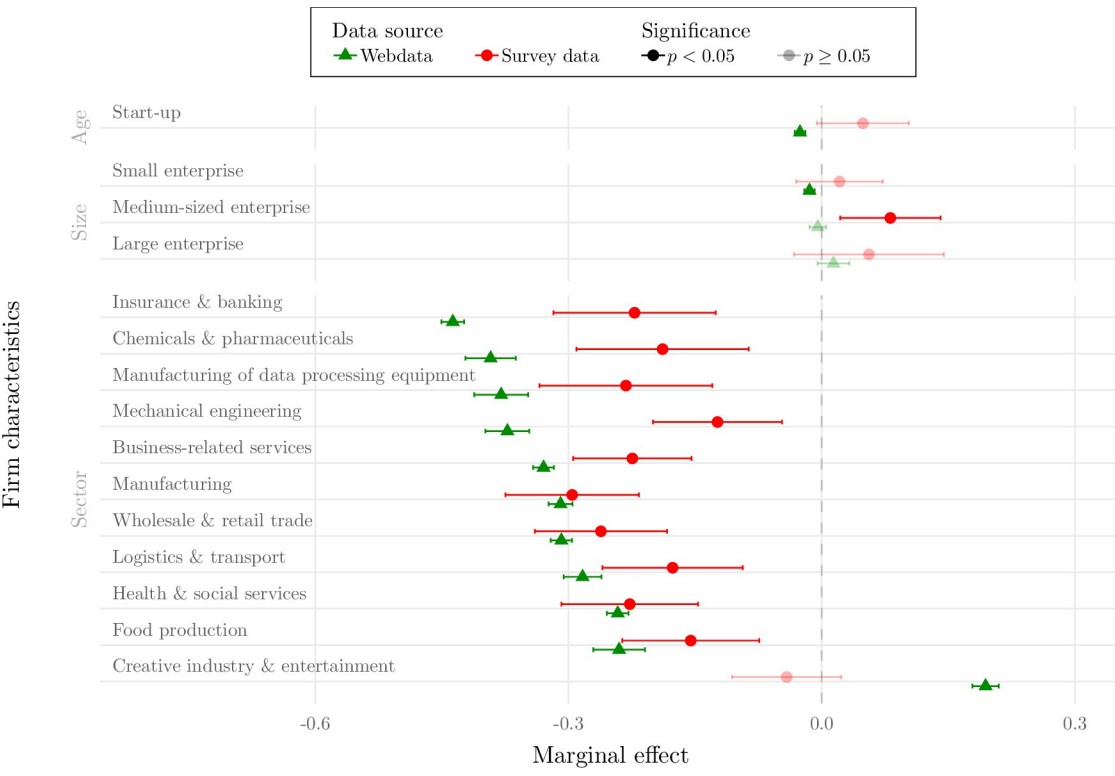

**Fig 6. Comparison webdata and survey data effect estimates.** Note: Figure shows average marginal effect estimates and corresponding 95%-confidence intervals of model 3 where the dependent variable (negative impact) is generated from webdata (green) and survey data (red). Dependent variable from webdata reflects whether the firm has reported a 'problem' reference on its corporate website in any of the web queries. Dependent variable from survey data refers to the question whether the firm has suffered negative impacts due to the pandemic in any of the three survey waves. Shaded estimates signal statistically insignificant effects at the 5% level. Incumbent firms (10 years and older) serve as baseline age group, micro-enterprises (number of employees $\leq$ 10) as baseline size group, accommodation and catering serves as baseline sector among the sector dummies. Marginal effects need to be interpreted relative to the baseline group(s).

The average marginal effects thus indicate by how many percentage points, on average, it is more likely that a firm with the respective characteristic is more likely/less likely affected by the pandemic. Given this interpretation of the regression results, four aspects are worth mentioning here: (i) it becomes apparent that, based on both webdata and survey data, age and size differences are modest at most and largely statistically insignificant in terms of their association with a negative crisis impact. However, the differences between economic sectors are substantial. Both regressions show that the probability of being negatively exposed to the shock is significantly lower in all sectors (with the exception of creative industries and entertainment) compared to the baseline sector accommodation and catering. (ii) The estimated effect directions are largely consistent between webdata and survey data and many of the estimated confidence intervals overlap. (iii) However, there are some exceptions. The most striking one being the differences between the average marginal effect estimate for creative industry and entertainment. While the survey-based results suggest that negative effects in the creative industry and entertainment sector are statistically no more likely than in accommodation and catering, the webdata-based results hint to a significant difference between the two sectors. According to our webdata-based results, creative industry firms are more likely affected by the exogenous shock as indicated by an estimated gap of close to 20 percentage points. Results in Section 3.3 based on credit rating changes indeed hint to slightly more adverse impacts in the creative industry and entertainment sector relative to pre-crisis rating downgrades, suggesting that the webdata effect estimate is reasonable. (iv) Due to the substantially higher observation number in the website-based dataset, the estimates' confidence bounds are much narrower compared to the ones of the survey estimates. The large-scale assessment that is possible with the large sample size from corporate websites is a clear advantage over relatively small-scale business surveys that often suffer high non-response rates. This has been show in Section 3.1, where the large sample size of the website data allows to assess impact heterogeneity across sectoral subgroups. Overall, it can be said that the effects derived from webdata closely resemble the effects derived from a traditional time and resource intensive business survey. This suggests that corporate communication data and the proposed way to generate indicators from it, is a useful instrument to learn the structural impact the pandemic had on the corporate sector. We see this is a useful way to overcome information deficits in order to better decide over economic countermeasures.

In a second analysis, we assess to which extent the context classes derived from the corporate website data serve as predictive indicators for later changes in a firms' credit rating. For this purpose, we regress firms' credit rating changes after June 01, 2020 on each of the five COVID-19 context classes generated in the first 'ad hoc' stage of our framework. Note that the context classes have been extracted from corporate websites between March 2020 and May 2020, i.e. *before* June 01, 2020. We express this time period with the index $\bar{t}$. $\Delta r_{i,\bar{t}+z}$ in model 4 refers to the first credit rating change of firm *i after* June 01, 2020 (with $z$ = number of days after $\bar{t}$; $\bar{t} + z$ = date of rating update). Finally, the regression incorporates the credit rating prior to the rating update which coincides with the firms' pre-crisis rating expressed via the index $\bar{t} - x$.

$$\Delta r_{i,\bar{t}+z} = \alpha + \beta_1 \text{Problem}_{i,\bar{t}} + \beta_2 \text{No problem}_{i,\bar{t}} + \beta_3 \text{Adaption}_{i,\bar{t}}$$
$$+ \beta_4 \text{Information}_{i,\bar{t}} + \beta_5 \text{Unclear}_{i,\bar{t}} + \gamma r_{i,\bar{t}-x} + \boldsymbol{\delta D}_i + \epsilon_i \tag{4}$$

with $\boldsymbol{D}$ as matrix comprising a collection of company age, size and sector controls.

Table 7 displays the regression estimates that result from this analysis. Regression specification (1) shows that the website categories have a significant leading indication concerning a firm's subsequent change in its credit rating. Looking at the sign estimate of the five categories,

**Table 7. Regression results: COVID-19 references on corporate websites as early indicators for changes in firm credit ratings.**

| | Regression specification | | | |
|---|---|---|---|---|
| | **(1)** | **(2)** | **(3)** | **(4)** |
| Problem$_i$ | 1.66*** | 1.68*** | 1.62*** | 0.42** |
| | (0.18) | (0.18) | (0.19) | (0.19) |
| No problem$_i$ | -1.70*** | -1.69*** | -1.73*** | -0.69 |
| | (0.42) | (0.42) | (0.43) | (0.43) |
| Adaption$_i$ | -0.46*** | -0.47*** | -0.33*** | -0.13 |
| | (0.08) | (0.08) | (0.08) | (0.08) |
| Information$_i$ | -0.24*** | -0.24*** | -0.23*** | -0.17*** |
| | (0.04) | (0.04) | (0.04) | (0.04) |
| Unclear$_i$ | -0.42*** | -0.42*** | -0.10 | -0.08 |
| | (0.12) | (0.12) | (0.12) | (0.12) |
| $r_{\bar{t}-x}$ | -0.09*** | -0.10*** | -0.11*** | -0.13*** |
| | ($< 0.01$) | ($< 0.01$) | ($< 0.01$) | ($< 0.01$) |
| Age controls | No | Yes | Yes | Yes |
| Size controls | No | No | Yes | Yes |
| Sector controls | No | No | No | Yes |
| N | 61,228 | 61,138 | 57,343 | 57,343 |

Note: Table shows Ordinary Least Squares (ols) estimates and white robust standard errors in parentheses for different regression specifications. Dependent variable, $\Delta r_{i,\bar{t}+z}$, is the change in a firm's credit rating after June 01, 2020. Main explanatory variables of interest are the web classes generated from the website text fragments (as count variables) in the early phase of the pandemic before June 01, 2020. Age, size and sector controls are analogous to the specifications in Fig 6. Significance levels:

*: $p < 0.10$,

**: $p < 0.05$,

***: $p < 0.01$

it becomes apparent that the webdata categories embody a predictive and meaningful indication concerning a firm's subsequent credit rating movement. In fact, firms which reported about problems in the context of COVID-19 on their websites suffered on average a statistically significant downgrade in their credit rating (positive sign reflects a deterioration in the firm's credit rating). Firms which have indicated that the pandemic is not causing problems on their business operations, by contrast, experienced a statistically significant upgrade on average (negative sign). Similarly, firms which have signaled adaption to the exogenous shock as well as such firms which only informed about COVID-19 in a broader context have also experienced upgrades on average, albeit at a lower magnitude. The same negative correlation is true if a Corona reference that could not be classified into a broader context were found on the company website. These results are robust when controlling for company age effects in specification (2), and additionally for firm size effects in specification (3). Both controls capture systematic differences in the exposure to economic shocks across firms of different size and age (such as the amount of cash reserves and collaterals for external financing). Interestingly, the statistical significance of the 'Unclear' category vanishes after controlling for company age and size which seems reasonable as the category is defined as not conveying context on the communicated effects of the pandemic. Ultimately, when adding sector fixed effects, which control for systematic differences in the pandemic's impact across sectors, in specification (4), it turns out that even within sectors the 'problem' class has still a leading indication on credit rating downgrades as indicated by the significant positive sign estimate. The same is true for the

'information' category, which still serves as significant leading indicator for later rating upgrades. For the remaining categories statistical significance vanishes when analyzing the forecasting power of the categories within sectors.

We see the results in this analysis as an important finding since they underpin that corporate website data serve as leading indicator of the pandemic's financial effects on corporations. Indeed, a credit rating downgrade has typically financial consequences for a firm as it impedes the company's ability to draw new credit lines due to its lower creditworthiness. In a phase of financial distress such as in the COVID-19 crisis, this increases the likelihood to end up in liquidity bottlenecks which may ultimately lead to financial insolvency. One problem of the sudden exogenous shock in the still ongoing COVID-19 crisis is that it has also pushed many companies with otherwise sound business models on the brink of financial solvency. From a policy perspective, this is undesirable and clearly called for quick policy support measures. In the early phase of the pandemic, the lack of information concerning the impacts on the corporate sector left policymakers little options but to grant subsidies as well as state-backed loans in a largely indiscriminate manner and at the cost of unprecedented net borrowing. Our results show that corporate website data and state-of-the art methods from the field of NLP bear the potential to cure this information deficit. With the early indication through 'ad hoc' web analyses, policymakers have a novel tool at hand that allows to detect structural distress in the economy early on. With our proposed framework, it is possible for policymakers to steer their response measures strategically to firms and sectors where help is required most urgently while not overburdening fiscal budget.

## 5 Conclusion

In this paper, we have presented a data-driven policy framework that not only provided policymakers with guidance for their economic support measures during the coronavirus pandemic, but also enabled them to capture the impact of the shock on the corporate sector at near real-time. Overall, the framework consists of three stages, with each stage, according to the timeliness of the data, allows for an impact assessment at different points in the course of the pandemic. These three stages, from an early stage 'ad hoc' web analysis using text fragments from company websites in the short run, to a differentiation of the various impacts via 'follow-up' business surveys in the mid-term, to 'retrospective' changes in firm's liquidity positions in the aftermath of the shock, show how information gaps that policymakers are confronted with in a highly dynamic situation can be successfully bridged. In this context, our results suggest that the classification of textual COVID-19 references found on company websites allows to generate meaningful impact categories which, in turn, reveal a strong heterogeneity of the pandemic's impact at fine granular industry level. The dynamic nature of website data made it possible to generate these insights immediately after the shock and at near real-time. In this vein, the classified Corona references strongly resemble the exposure results that are obtained via traditional business surveys, with the difference that the survey results have only become available several weeks after the shock had hit the economy. Moreover, we show that the classified text fragments serve as leading indicators in predicting credit rating downgrades of firms that are adversely affected by the economic shock. These insights pose a valuable update to policymakers' information set and provide empirical evidence to justify swift and targeted response measures.

The early stage assessment via COVID-19 references extracted for a large sample of corporate websites is a novel and promising approach that shows how alternative sources of unstructured online data and methods from the field of NLP can create insights for policymakers when traditional sources of data are only available with non-negligible time delay. The

coronavirus pandemic has shown that in situations where policymakers need to respond quickly, but information deficits make it barely possible to determine where government assistance is channeled most efficiently, public aid measures are largely granted on a lump-sum basis. In fact, in Germany, this information deficit has led the Ministry of Finance to choose the 'bazooka' [52] instead of well-dosed and targeted liquidity injections as instrument to support companies in the early phase of the pandemic. Our framework is designed to help overcome information deficits that lead to otherwise undifferentiated support measures. In this context, we see this study as a first step towards real-time decision support for economic policymakers. Given further research and development, we argue that our framework can serve as a monitoring framework applicable to a wider range of economic shocks.

There are, however, limits to our analysis. First and foremost, not all companies have their own corporate website domain, which likely biases our web-based analysis results. Previous studies have shown that URL coverage of German companies is at 46% [32]. Especially among smaller firms the fraction without corporate URL is comparatively high. However, this does not necessarily mean that these companies do not have a corporate online presence at all. Often small and micro firms host corporate profiles on social media platforms to communicate with their stakeholders. It requires further research to detect, access and analyze these online presences to acquire an even more complete picture of corporate communication on the internet in times of economic shocks. Next, company website content is essentially self-reported information that generally bears the risk that firms communicate their current situation overly optimistic (or pessimistic). Interestingly, this study has revealed that in times of economic crises this does not seem to be necessarily the case. On the contrary, we find that close to 70,000 firms reported about problems that they are facing in relation to the pandemic. This equals 35% of all firms that published COVID-19 references on their websites and is substantial given the potential consequences of communicating 'problems' to such a broad audience.

If machine learning-based analysis systems, such as the framework we have presented, indeed find their way into the standard indicator toolkit of policymakers, the question of interpretable (and fair) prediction results will also arise. Complex machine learning models in particular are often deemed as difficult to understand 'black boxes' that do not allow any clear conclusions to be drawn about the factors that are ultimately decisive for predictions and forecasts. In the near future, frameworks like ours will have to integrate aspects of *explainable AI* (see for example Barredo Arrieta et al. [53]) in order to provide decision-makers not only with reliable, but also explainable information as a basis for making informed decisions.

Despite the theoretical drawbacks of our proposed framework, we believe that it is a useful research contribution towards policy guidance that balances timeliness, depth and costs of different data sources. Especially in times of crisis, when sudden shocks cause major disruptions, exploring alternative sources of data is critical to provide timely insights to decision-makers. In this regard, we believe that webdata and other real-time online sources not only serves as a tool to capture business impacts in highly dynamic situations, but also has the potential to support policymakers across a broader spectrum. It is left to future research to explore the value of webdata for policy on a larger scale.

## Supporting information

**S1 Table. Search terms for querying COVID-19 references on corporate websites.** Searches were conducted case insensitive. Spaces in the search terms were treated as wildcards where any two characters instead of the space also led to a match. In this way, we allowed a greater degree of variation in the search for Corona references.
(PDF)

**S2 Table. Fraction of firms with COVID-19 references on corporate websites.** Table shows the fraction (in %) of companies within the presented sector-size strata where we could find COVID-19 references on the corporate website in at least one of our web queries. Fractions reveal that larger firms are more likely to report about the virus on their websites. The numbers also show great heterogeneity across sectors. The last column presents the sample size of corporate website addresses across sectors.
(PDF)

**S3 Table. Mapping EU NACE Revision 2 divisions to sector groups.** Table shows the translation of EU's NACE Revision 2 divisions [47] into the sector groupings used in this study.
(PDF)

**S4 Table. Mapping firm characteristics to size group.** Table shows translation of firm characteristics into company size classes as defined by [54] and also used in this study.
(PDF)

**S5 Table. Examples of COVID-19 references found on corporate websites.** Table shows three website text examples for each of the five context classes retrieved from distinct corporate websites.
(PDF)

**S6 Table. COVID-19 website-generated impact values at sector level.** Table shows sector level impact values as displayed in Fig 3. Impact values are defined as the proportion of companies that communicated about the pandemic in the respective context within that sector. The unweighted average of the impact values across all sectors forms the grey-shaded reference area in Fig 3.
(PDF)

**S7 Table. Effect heterogeneity within wholesale sector.** Table shows impact values of wholesale companies disaggregated by NACE Revision 2 classes (4-digit-level) [47]. Impact values are defined as the proportion of companies that communicated about pandemic-related problems within the respective subsector. *N* refers to the number of observations in the subsector.
(PDF)

**S1 Fig. Effect heterogeneity across geographic regions.** Figure shows regional impact values to demonstrate the presented framework's capability to monitor regional hotspots where comparatively many companies are negatively affected by the shock. Impact values are presented for three selected web queries in March, April and May 2020. Regional impact values show at the beginning of the pandemic strong problem reporting of companies located in cross-border regions. Investigation of the text references showed that these values were driven by specialized companies located at transportation hubs that were virtually unused during the lockdown. Towards the end of the first economic shutdown beginning of May, problem reports diminished. Impact values are defined as the proportion of companies that reported about pandemic-related problems within that region.
(TIF)

**S1 Appendix. Survey details.** The business surveys presented in this study are the result of a joint research project between the polling agency KANTAR and ZEW—Leibniz Centre for European Economic Research funded by the German Federal Ministry for Economic Affairs and Energy (BMWi). Scope of the project was to provide early insights on the effects of the pandemic on the German business sector based on website analyses and survey data. The sampling strategy followed a stratified random sample drawn from the MUP which comprises the

near universe of active firms in Germany [37]. Stratification was conducted by industry (see S3 Table) and employee size classes (see S4 Table) and ensured sufficient regional coverage across federal states. Computer aided telephone interviews following a predefined stratification matrix by size classes and industries were conducted. The predefined stratification matrix formed the basis for gross sampling as well as for sampling management during fieldwork. A target of at least $N = 30$ interviews in the industries ensured that sufficient observations were available at the sector-level to ensure credible conclusions for all sectors as it is done in this study. Overall, three recurring survey waves over the period from April to September 2020 have been conducted. Not all firms agreed upon processing their responses beyond the scope of the aforementioned research project. These firms are excluded from the analyses in this paper. The table below provides further details concerning the total number of interviewed companies ($N_{overall}$) and the number of firms included in this study ($N$) for each of the survey waves. As far as possible, companies were continuously surveyed in all three survey waves. If companies refused to participate again or could not be reached in a subsequent wave, new companies were drawn from the stratified gross sample in order to meet the target observation number of the respective survey wave.
(PDF)

## Acknowledgments

We are grateful for valuable comments from the participants of the Conference on New Techniques & Technologies for Statistics (NTTS2021) organized by Eurostat. Moreover, we are indebted to our anonymous reviewers for comments and criticism that greatly improved the manuscript. Tobias Weih and Sabrina Pataky provided excellent research assistance.

## Author Contributions

**Conceptualization:** Julian Oliver Dörr, Jan Kinne, David Lenz, Georg Licht, Peter Winker.

**Data curation:** Julian Oliver Dörr, Jan Kinne, David Lenz.

**Formal analysis:** Julian Oliver Dörr, Jan Kinne, David Lenz.

**Funding acquisition:** Georg Licht, Peter Winker.

**Investigation:** Julian Oliver Dörr, Jan Kinne, David Lenz.

**Methodology:** Julian Oliver Dörr, Jan Kinne, David Lenz.

**Project administration:** Peter Winker.

**Resources:** Georg Licht.

**Software:** Julian Oliver Dörr, Jan Kinne, David Lenz.

**Supervision:** Julian Oliver Dörr, Jan Kinne, David Lenz.

**Validation:** Julian Oliver Dörr, Jan Kinne, David Lenz.

**Visualization:** Julian Oliver Dörr, Jan Kinne, David Lenz.

**Writing – original draft:** Julian Oliver Dörr, Jan Kinne, David Lenz.

**Writing – review & editing:** Julian Oliver Dörr, Jan Kinne, David Lenz, Georg Licht, Peter Winker.

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
