## [Decision Letter · Decision Letter 0]

8 Nov 2021

PONE-D-21-23221Combining big and small data: An integrated data framework for policy guidance in times of dynamic economic shocksPLOS ONE

Dear Dr. Dörr,

Thank you for submitting your manuscript to PLOS ONE. After careful consideration, we feel that it has merit but does not fully meet PLOS ONE’s publication criteria as it currently stands. Therefore, we invite you to submit a revised version of the manuscript that addresses the points raised during the review process.

Several aspects of the paper should be clarified, as also mentioned by the reviewers. The performance of the deep learning classifier should be evaluated using the recognized metrics (Precision, Recall, etc.). The usefulness of the approach should be better demonstrated, as also highlighted by one of the reviewers.

We look forward to receiving your revised manuscript.

Kind regards,

Liviu-Adrian Cotfas

Academic Editor

PLOS ONE

Journal Requirements:

Reviewers' comments:

Reviewer's Responses to Questions

**Comments to the Author**

1. Is the manuscript technically sound, and do the data support the conclusions?

Reviewer #1: Partly

Reviewer #2: Partly

2. Has the statistical analysis been performed appropriately and rigorously? 

Reviewer #1: No

Reviewer #2: Yes

3. Have the authors made all data underlying the findings in their manuscript fully available?

Reviewer #1: No

Reviewer #2: Yes

4. Is the manuscript presented in an intelligible fashion and written in standard English?

Reviewer #1: Yes

Reviewer #2: Yes

5. Review Comments to the Author

Reviewer #1: This work introduces an integrated data framework for guiding policymakers during a crisis in a timely and cost-effective manner.

The core of this work focuses on finding different strategies to analyze various data sources from companies that can provide the bigger picture during the Covid-19 pandemic, whether it's in an early stage near-real-time, in a follow-up, or a retrospective stage (in the aftermath of the shock).

The authors conclude that their framework provides policymakers with guidance for their economic support measures in times of sudden shocks.

Strengths

*) This work provides an interesting study of the economic sector struggles during Covid-19.

*) The idea to combine data from different data sources (e.g., online, surveys, official data) to aid in forecasting is very good, although not new.

*) Most of the references are new and relevant.

Weaknesses

-----------------

*) The title is more generic than what the article presents. The authors should deliver a "data framework for policy guidance in times of dynamic economic shocks" but the solution is very specific to the Covid-19 scenario. It will be very hard to generalize (to other types of crisis) as most of the steps are done manually and not automatically, e.g., they somehow chose some specific keywords /terms to search in the text (section 3.1) and some classes for the classifier. Something like automated detection of the keywords would make the solution more general. Even the evaluation of the model (XLM-RoBERTa) was done only manually, instead of also using specific evaluation metrics. They did not discuss the precision, accuracy, or recall of the model.

*) The theoretical and experimental analyses of the paper are not rigorous.

*) By only analyzing the Covid-19 scenario, they do not demonstrate that their solution is efficient and optimal in a general crisis situation. Also, at the start of the lockdown was very clear what sectors are going to suffer the most. I am not sure what in-depth insights this study offers.

To sum up, the primary issue with this paper is that the contribution as a piece of research is not clear.

Observations

--------------

- the figures and some of the tables were not added in the article, they can be found at the end of the document (all the figures) or can be downloaded separately (tables Si_Table).

Reviewer #2: The paper proposes a framework for guiding policy makers in the context of a major economic event by analyzing small and big data.

The topic is interesting and worth investigating, given the recent COVID-19 economic shock. The paper is well structured, includes a comprehensive enough literature review and uses recent transformer-based language models. However, many aspects of the paper are described in rather vague terms.

Additional details should be provided regarding the classifier used for categorizing the announcements on the companies’ websites. The current description can be considered quite vague. For example, the paper should state how many of the 4,347 manually classified text passages belonged to each considered category. Has the dataset been a balanced or an unbalanced one? How have the authors handled the training process if the training dataset has been unbalanced? Additionally, what happened when a disagreement between the two annotators has been encountered? The paper should evaluate the performance of the classifier used for using the standard metrics such as Precision, Recall, Accuracy and F1-Score.

Numerical values should also be provided in the text discussing Figure 3. The authors could also choose to mention the values in a table.

Precise information regarding the survey should also be included. The current version of the paper does not for example mention the exact number of companies that have completed the survey in the text of the paper (only included in Table 1).

6. PLOS authors have the option to publish the peer review history of their article (what does this mean?). If published, this will include your full peer review and any attached files.

Reviewer #1: No

Reviewer #2: No

---

## [Author Response · Author response to Decision Letter 0]

23 Dec 2021

Dear Reviewers,

We are grateful to you for thoroughly reading our paper and for constructive comments. After going through each of the suggestions, we invested every effort to incorporate all of them. This contributed, we hope, to elaborate an improved version of the paper. Where it was not possible to address the comments in a way proposed by the reviewers, we did our best to provide a convincing explanation.

We provided answers to comments respecting the order followed by the reviewers. Where different issues were included in a single comment, we separated the paragraphs in order to provide punctual answers to each of them. We used alphabetic letters to identify all the comments and we wrote our responses in italics. We have uploaded the point-by-point responses to the comments as PDF. New parts in the revised version of the paper are marked in blue.

Let us once again express our gratitude for your insightful comments and suggestions.

Best regards,

The Authors

---

## [Editor Report · Decision Letter 1]

31 Jan 2022

An integrated data framework for policy guidance during the coronavirus pandemic: Towards real-time decision support for economic policymakers

PONE-D-21-23221R1

Dear Dr. Dörr,

We’re pleased to inform you that your manuscript has been judged scientifically suitable for publication and will be formally accepted for publication once it meets all outstanding technical requirements.

Kind regards,

Liviu-Adrian Cotfas

Academic Editor

PLOS ONE
---

## [Editor Report · Acceptance letter]

4 Feb 2022

PONE-D-21-23221R1 

An integrated data framework for policy guidance during the coronavirus pandemic: Towards real-time decision support for economic policymakers 

Dear Dr. Dörr:

I'm pleased to inform you that your manuscript has been deemed suitable for publication in PLOS ONE. Congratulations! Your manuscript is now with our production department. 

Kind regards, 

on behalf of

Dr. Liviu-Adrian Cotfas 

Academic Editor

PLOS ONE